# Blast from the past II: Constraints on heavy neutral leptons from the BEBC WA66 beam dump experiment

Ryan Barouki, Giacomo Marocco* & Subir Sarkar

Rudolf Peierls Centre for Theoretical Physics, University of Oxford
Parks Road, Oxford OX1 3PU, United Kingdom
(*giacomo.marocco@physics.ox.ac.uk)

November 4, 2022

## Abstract

We revisit the search for heavy neutral leptons with the Big European Bubble Chamber in the 1982 proton beam dump experiment at CERN, focussing on those heavier than the kaon and mixing only with the tau neutrino, as these are far less constrained than their counterparts with smaller mass or other mixings. Recasting the previous search in terms of this model and including additional production and decay channels yields the strongest bounds to date, up to the tau mass. This applies also to our updated bounds on the mixing of heavy neutral leptons with the electron neutrino.

## 1   Introduction

Neutrinos have small but non-zero masses, the origin of which is unknown. An attractive explanation involves extending the Standard Model (SM) by adding to it right-handed neutrinos, thus generating small masses for the left-handed neutrinos. The most popular is the 'seesaw' mechanism for Majorana masses which has many variants [1]. Such models have open parameter space where the heavy neutral leptons (HNL) mix only with a single

flavour of active neutrinos [2]. Our goal in this work is to bound the currently least constrained possibility of mixing between HNLs and the tau neutrino. In this simple model, an HNL $N$ has a mass $m_N$ and mixes with the $\nu_\tau$ with a strength given by $U_{\tau N}$. This mixing arises from one of the few renormalisable operators — the so-called neutrino portal — that may consistently be added to the Standard Model to couple it to a 'dark sector', so is a promising target in the search for new physics beyond the electroweak scale [3]. Our bounds also apply to neutrino portal dark sector models where the HNL is a Dirac fermion [4]

Many constraints exist on the $m_N - U_{\tau N}$ parameter space — see [5] for a comprehensive review and discussion of proposed experiments at the LHC Forward Physics Facility (FPF). Below the kaon mass, the ND280 detector at T2K places strong bounds [6]. The ArgoNeuT experiment [7] probed HNL masses around a GeV, but better bounds may be extracted from $B$ factories [8], in particular BABAR [9] and Belle [10]. The strongest constraint comes however from reanalysis [11–13] of the bound from the CHARM beam dump experiment [14]. At higher masses, DELPHI at LEP constrained HNLs that may be produced in $Z$-decays [15] while ATLAS [16] used $W$-decays. Complementing laboratory experiments, arguments based on big bang nucleosynthesis [17] are relevant in regions of parameter space where the HNLs are long-lived [18, 19].[1] Since the experimental bounds are less restrictive above the kaon mass [21], we focus here on $m_N \sim 0.5 - 1.8$ GeV.

Data from the Big European Bubble Chamber (BEBC) WA66 experiment in the 1982 CERN beam dump (400 GeV protons from the SPS) [22] had been used to carry out a dedicated search for HNLs [23] contemporaneously with CHARM. However that analysis focussed on HNL production (and decay) via mixing with electron and muon neutrinos; the production of HNLs in $\tau$ decays was not considered nor were decays via neutral currents taken into account. Given that BEBC continues to set world-leading bounds on other new physics such as dark photons [24], magnetic moments, and millicharged particles [25], we reassess its sensitivity to HNLs mixing with $\nu_\tau$, addressing the above lacunae. We also carry out a reanalysis of HNL mixing with $\nu_e$ in order to include all relevant decay modes and correct a decay rate in [23] that omitted an interference contribution, thus obtaining a more restrictive bound on the mixing angle. The bounds from BEBC [23] have not been noted in many otherwise comprehensive recent discussions on HNLs e.g. [26–29].

## 2 Heavy neutrino production

The flux of HNLs produced in the beam dump is specified by the absolute number of HNLs produced, $\mathcal{N}_N$, as well as their differential distribution, $\mathrm{d}^3\mathcal{N}_N$, in energy and momentum. We first calculate the latter.

For a production channel labelled by $i$, once the initial distribution $\mathrm{d}^3\sigma_i$ of the parent and the decay distribution $\mathrm{d}^3\mathcal{N}_i$ of the child are known, the child's distribution in the lab frame is readily obtained. It is necessary to integrate over these distributions on the subspace where the boost brings the cms momentum to the lab frame momentum in

---

[1]Combining the laboratory limits on the mixing of neutrinos from BEBC, CHARM etc with cosmological bounds on their lifetime was first done in [20] in order to complete exclude a $\nu_\tau$ with mass $> 2m_e$.

question, i.e.,

$$\frac{\mathrm{d}^3 \mathcal{N}_N}{\mathrm{d}E_\chi \mathrm{d}\cos\theta_\chi \mathrm{d}\phi_\chi} = \sum_i \int \mathrm{d}E^* \mathrm{d}\phi^* \mathrm{d}\cos\theta^* \int \mathrm{d}E \mathrm{d}\phi \mathrm{d}\cos\theta \frac{\mathrm{d}^3\sigma_i}{\mathrm{d}E\mathrm{d}\phi\mathrm{d}\cos\theta} \cdot \frac{\mathrm{d}^3 \mathcal{N}_i}{\mathrm{d}E^*\mathrm{d}\phi^*\mathrm{d}\cos\theta^*} \cdot$$
$$\delta(E_\chi - E')\delta(\cos\theta_\chi - \cos\theta')\delta(\phi_\chi - \phi'), \tag{1}$$

where $E, \phi, \theta$ are, respectively, the parent energy, azimuthal angle and polar angle in the lab frame, the starred quantities describe the child in the cms frame, while the primed quantities are obtained by boosting the starred quantities by the Lorentz transformation associated with the unstarred ones. This integral cannot be done analytically, so we obtain the distribution by sampling the underlying $\mathrm{d}^3\sigma, \mathrm{d}^3\mathcal{N}$ distributions and explicitly performing the boosts. Furthermore, since the parent particles are focused predominantly along the beam axis, we have $\phi \simeq \phi'$, hence the $\phi$ integral is trivial.

Turning now to the absolute number of produced HNLs, this depends on the normalisations of $\mathrm{d}^3\sigma_i$ and $\mathrm{d}^3\mathcal{N}_i$. To eliminate systematic errors in the extraction of these quantities, associated e.g. with the adopted model of proton-nucleon interactions in the beam dump, we calibrate this directly using the *concommitant* flux of active neutrinos. This was measured at BEBC [22], and is consistent with their dominant source being the three-body prompt decays of $D^\pm$ and $D^0$ mesons [23]. Hence the total number of HNLs produced, $\mathcal{N}_N$, can be directly related to the total number of ($\sim$massless) active neutrinos of a particular species $\mathcal{N}_{\nu_\ell}$ via

$$\frac{\mathcal{N}_N}{\mathcal{N}_{\nu_\ell}} \simeq \frac{\sum_i \sigma(pN \to P_i + X)\mathrm{Br}(P_i \to N + Y)}{\sigma(pN \to D^+D^- + X)\mathrm{Br}(D^\pm \to \ell\nu_\ell + X) + \sigma(pN \to D^0\bar{D}^0 + X)\mathrm{Br}(D^0 \to \ell\nu_\ell + X)}, \tag{2}$$

where we sum over all parent particles $P_i$ that produce HNLs in their decays. We take $4\sigma(pN \to D_s + X) = 2\sigma(pN \to D^+D^- + X) = \sigma(pN \to D^0\bar{D}^0 + X)$ in accordance with data from the Fermilab E769 experiment [30], so that all cross-sections in the denominator above are proportional to each other. If all the production cross sections $\sigma(pN \to X)$ in the numerator too are proportional (to be justified when we identify the $P_i$ that appear in this equation), then the hadronic dependence drops out *modulo* the proportionality constants, thus simplifying the calculation considerably and yielding a robust constraint. In the WA66 experiment, it was estimated that $4.1 \times 10^{-4}$ muon neutrinos were produced via $D$ decays per proton on target [22], which allows for direct calculation of $\mathcal{N}_{\nu_\ell}$. The above procedure minimises systematic uncertainties in the overall flux normalisation when the angular distribution is known.

We have thus reduced the problem to obtaining initial parent and final decay distributions, as well as branching ratios for certain reactions. These depend on the specific production modes in question, which we now delineate.

## 2.1 HNL production through decays

The particle decays that generate HNLs depend on which active neutrino it mixes with. For decays mediated by the neutral current, flavour is conserved (up to the $N - \nu_\ell$ mixing), and so the HNL appears in conjunction with an associated lepton. In particular, for mixing with $\nu_\tau$, if $m_N > m_P - m_\tau$, then HNL production from decay of a parent of mass $m_P$ is kinematically forbidden. We can now sort the production channels by those allowed with a non-zero $U_{\tau N}$, before moving on to the non-zero $U_{eN}$ case.

### 2.1.1 Mixing with $\nu_\tau$

HNLs above the kaon mass cannot be produced in the decays of mesons containing just the lightest four quarks. In principle they may be produced in the decays of $B$ mesons, but their production cross-section is only $\sim 1$ nb at 400 GeV [31,32], which is too small to yield any interesting constraints. Hence the dominant contribution to HNLs above the kaon mass that mix solely with the tau neutrino is from tau lepton decay. The taus are themselves produced in $D$ decays, so we may apply Eq.(1) to determine their flux, which in turn determines the HNL flux from their subsequent decay.[2] Note that since the $\tau$ ultimately comes from a $D$, its production probability is proportional to the *same* hadronic cross-section, as was anticipated above in the discussion following Eq.(2).

The $D_s$ meson is the dominant source of $\tau$ leptons, which in turn decay to heavy HNLs. We thus need their differential distribution which is usually parameterised as [33]:

$$\frac{\mathrm{d}^2\sigma}{\mathrm{d}x_\mathrm{F}\mathrm{d}p_\mathrm{T}^2} \propto \mathrm{e}^{-bp_\mathrm{T}^2}(1 - |x_\mathrm{F}|)^n, \tag{3}$$

where $x_\mathrm{F} = 2p_L^{\mathrm{CM}}/\sqrt{s}$ is twice the longitudinal momentum in the cms frame (relative to the cms energy), $p_\mathrm{T}$ is the transverse momentum and the parameters $b$ and $n$ must be extracted from data. While $b$ can be considered to be independent of both the cms energy and the quark content of the charmed meson [34], $n$ may in general depend on both of these quantities. In the absence of specific data, we take $n$ for $D_s$ production to be the same as for $D^0$, $D^\pm$ mesons. To parameterise the production in the WA66 experiment, we use the results from the WA82 experiment [35], since both experiments used the same target material (copper), as well as similar beam energies (370 GeV for WA82 *cf.* 400 GeV for WA66). We adopt $b = 0.93 \pm 0.09$ GeV$^{-2}$ and $n = 6.0 \pm 0.3$ [35]; somewhat different values for $n$ were quoted by other experiments e.g. [36,37], but this is not as important for HNL production as the transverse momentum distribution which is set by $b$.

The tau has three main decay modes which result in an HNL, $\tau^\pm \to \pi^\pm N$, $\tau^\pm \to \rho^\pm N$ and $\tau \to \ell\nu_\ell N$. The two-body decays are determined simply by energy-momentum conservation, with branching ratio [38]:

$$\mathrm{Br}(\tau \to \pi N) = \mathrm{Br}(\tau \to \pi\nu_\tau) \cdot \sqrt{\lambda(y_\pi, y_N)}g(y_\pi, y_N)|U_{\tau N}|^2,$$
$$\lambda(x,y) = \frac{1 + x^2 + y^2 - 2(x + y + xy)}{(1-x)^2},$$
$$g(x,y) = \frac{(1-y)^2 - x(1+y)}{1-x}, \tag{4}$$
$$y_\pi \equiv \left(\frac{m_\pi}{m_\tau}\right)^2, \quad y_N \equiv \left(\frac{m_N}{m_\tau}\right)^2;$$

and

$$\mathrm{Br}(\tau \to \rho N) = \mathrm{Br}(\tau \to \rho\nu_\tau) \cdot \sqrt{\lambda(y_\rho, y_N)}g'(y_\rho, y_N)|U_{\tau N}|^2,$$
$$g'(x,y) = \frac{(1-y)^2 + x(1 + y - 2x)}{1 + x - 2x^2}, \tag{5}$$
$$y_\rho \equiv \left(\frac{m_\rho}{m_\tau}\right)^2.$$

We take $m_\pi = 140$ MeV, $m_\rho = 770$ MeV, $m_\tau = 1777$ MeV, and $\mathrm{Br}(\tau \to \pi\nu_\tau) = 0.108$, $\mathrm{Br}(\tau \to \rho\nu_\tau) = 0.252$ [39].

---

[2]We make the simplification of averaging over the $\tau$ spin, so that the final decay is isotropic in its rest frame, as it is for the scalar mesons.

By contrast, the distribution in a three-body decay depends on the mediator, which is here the $W$ boson. The angular distribution of the decays is dictated by the orientation of the $\tau$ polarisation, which is in turn set by the chiral structure of the weak interactions. However, we may average over this effect taking into account the equal number of $D_s$ and $\bar{D}_s$ which produce the ensemble of polarised leptons. The decays are thus fully characterised by their energy distribution in the $\tau$ rest frame, which is approximated at low momentum transfer $q^2 \ll M_W^2$ by [40–42]

$$\frac{d\Gamma}{dx} = \Gamma_0 x^2 \beta \left( 3 - 2x + \frac{x}{4}(3x - 4)(1 - \beta^2) \right), \tag{6}$$

where $x = 2E_N/m_\tau$ is twice the energy fraction carried by the HNL, $\beta = \sqrt{1 - (m_N/E_N)^2}$, and the normalisation $\Gamma_0$ is set by the observed active neutrino flux. The mass of the lepton pair has been neglected above but we keep explicit the dependence on the possibly sizeable HNL mass $m_N$. The energy fraction $x$ may take values between $x_{\min} = 2m_N/m_\tau$ and $x_{\max} = 1 + (m_N/m_\tau)^2$. The normalisation is given by [43]

$$\mathrm{Br}(\tau \to \ell\nu_\ell N) = \mathrm{Br}(\tau \to \ell\nu_\ell\nu_\tau) \cdot f(y_N)|U_{\tau N}|^2,$$
$$f(x) = 1 - 8x + 8x^3 - x^4 - 12x^2 \log x. \tag{7}$$

The above equations fully determine the hypothetical HNL flux in terms of the observed active neutrino flux. As previously stated, the total number of HNLs may be obtained by a simple rescaling of the number of ($\sim$massless) neutrinos observed, while the differential distribution is obtained from a full Monte Carlo. The results of this are shown in Fig.1 which displays the angle of HNLs produced with respect to the beam axis, against the energy of the HNL; we plot $\theta_N^2$ rather than $\theta_N$ since the isotropic measure on the sphere is flat in the former quantity. Note that as expected the HNLs are highly focused along the beam axis due to the large Lorentz factors of their parent particles.

### 2.1.2 Mixing with $\nu_e$

HNLs that mix with electron (or muon) neutrinos are produced readily in beam dumps, as the abundant heavy charmed mesons create them in their decays. We consider only two-body decays, as these are the dominant source of massive HNLs, unlike $\sim$massless active neutrinos $\nu_\ell$ that are subject to helicity suppression, e.g. [38,44,45]. The branching ratio for these processes is, in ratio to the corresponding SM branching ratio:

$$\mathcal{R} = \mathrm{Br}(D^\pm \to \ell N)/\mathrm{BR}(D^\pm \to \ell\nu_\ell) = \lambda^{1/2}(x_\ell, x_N)h(x_\ell, x_N)|U_{\tau N}|^2,$$
$$\text{where,} \quad \lambda(x,y) = \frac{1 + x^2 + y^2 - 2(x + y + xy)}{(1-x)^2}, \quad h(x,y) = \frac{x + y - (x-y)^2}{x(1-x)},$$
$$x_\ell = \left(\frac{m_\ell}{m_D}\right)^2, \quad x_N = \left(\frac{m_N}{m_D}\right)^2. \tag{8}$$

The behaviour of the above ratio is shown in Fig.3. Since this is a two-body decay, the differential distributions are fully determined by conservation laws.

## 3 Heavy neutrino decay and detection

In order to be detected, the HNLs produced in the beam dump must reach the detector and then decay within it. The probability for the HNL to reach the detector depends on all the possible detection channels open to it, as well as the mediating interactions. For

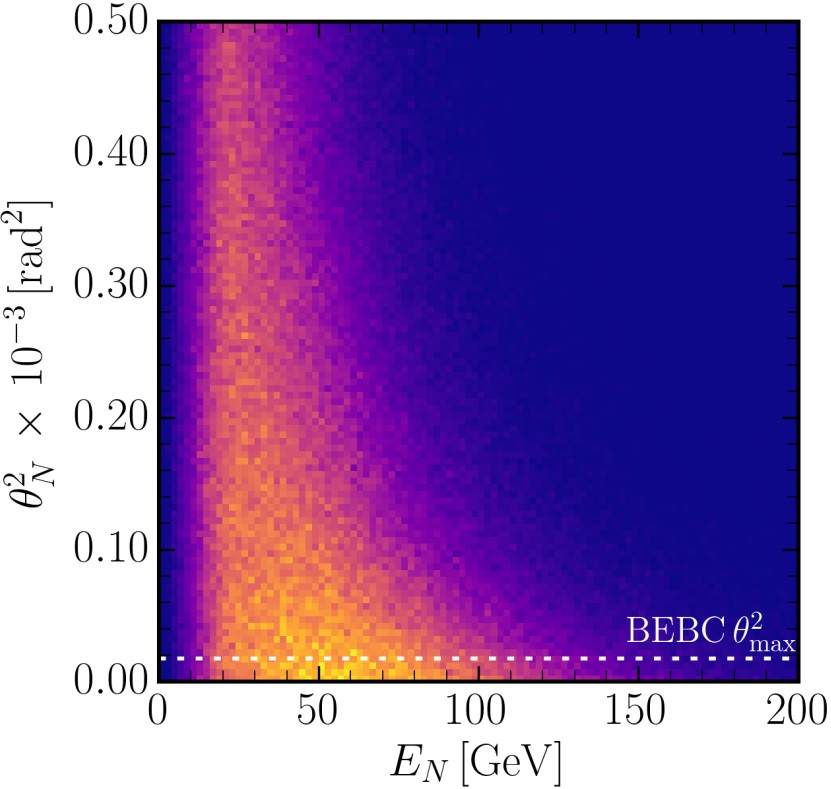

Figure 1: A 2D histogram of the HNL energy distribution, for $m_N = 1$ GeV, against the square of the angle with respect to the beam axis. We consider HNLs that mix only with $\nu_\tau$s, so are produced solely from $\tau$ decays. The dashed horizontal line indicates the opening angle of BEBC as seen from the production point in the beam dump.

simplicity, we consider only SM particles in the final state, i.e. HNL decay via the known electroweak bosons. Our analysis is easily generalised to decays via other mediators, see e.g. [46, 47]. The probability $P$ for an HNL to reach the detector at a distance $L'$ from the target and then decay within the length $L \ll L'$ of the detector is:

$$P = \exp\left(-\frac{m_N L' \Gamma}{p_N}\right)\left[1 - \exp\left(-\frac{m_N L \Gamma}{p_N}\right)\right], \qquad (9)$$

where $p_N$ is the momentum of an HNL, and $\Gamma$ is the total decay rate. In the small mixing regime, where $L' m_N \Gamma / p_N \ll 1$, we can linearise this to write

$$P \simeq 1.5 \times 10^{-8} \left(\frac{L}{1\,\mathrm{m}}\right) \cdot \left(\frac{100\,\mathrm{GeV}}{p_N}\right)\left(\frac{m_N}{1\,\mathrm{GeV}}\right)^6 \cdot \left(\frac{|U_{\tau N}|^2}{10^{-7}}\right), \qquad (10)$$

to illustrate a benchmark decay rate for a purely leptonic electroweak decay. Note that the mixing angle factorises, simplifying the Monte-Carlo simulations substantially. In this small-mixing regime, we may place an upper bound on the size of the mixing angle; however, for larger mixings, we can instead place a lower bound by requiring that the HNLs decay *before* they can reach the detector.

The detection probability depends solely on the decay channels for which a search was carried out in BEBC, and is associated with an experimental efficiency $\epsilon$. The number of

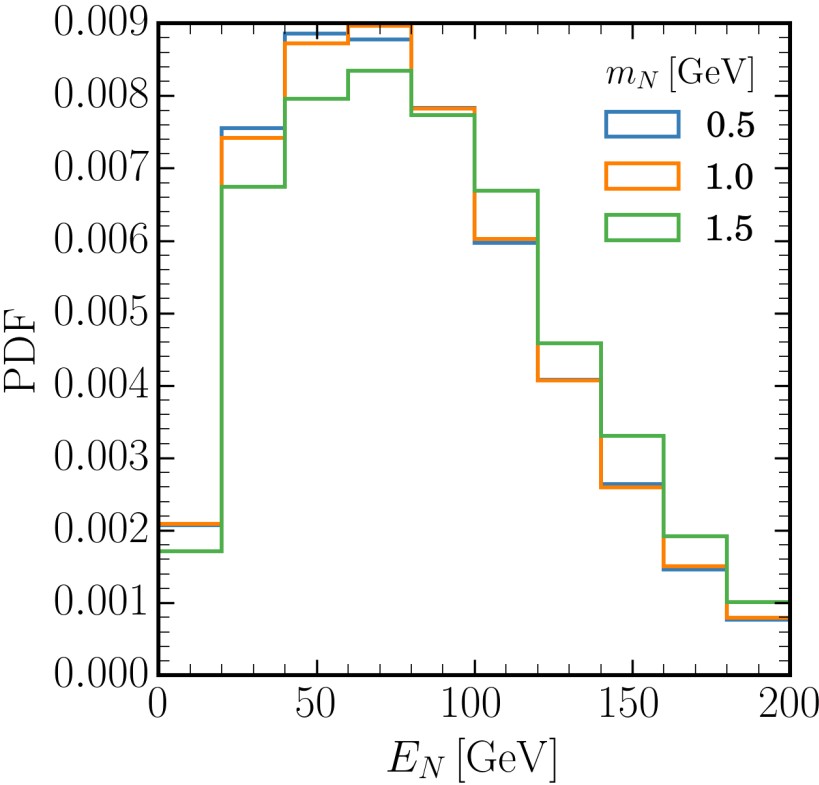

Figure 2: A histogram of the energy distribution of HNLs produced within the BEBC opening angle. Results are shown for three values of the HNL mass $m_N$ to illustrate the insensitivity to the mass until it approaches the production threshold of $m_\tau$.

observed events is related to the number of HNLs produced in the beam dump as

$$\mathcal{N} = \mathcal{N}_N \, \Omega \, \langle P \rangle_\Omega \sum_\alpha \frac{\Gamma_\alpha}{\Gamma} \cdot \epsilon_\alpha, \tag{11}$$

where $\mathcal{N}_N$ is given by Eq.(2), $\Omega$ is the geometric acceptance set by the solid angle subtended by the detector and $\langle \cdot \rangle_\Omega$ indicates an average over HNLs that lie within this acceptance, while the sum is over experiment-specific channels. The efficiency $\epsilon$ is a combination of factors which depends on both the detector response and the HNL decay channel. At BEBC, searches were made for $\ell^- \pi^+ / \ell^+ \pi^-$ and $\ell^- \ell^+ \nu$ where $\ell = e, \mu$ [23]. HNL decay candidates were required to have an oppositely charged particle pair (with momentum $> 1$ GeV/$c$ for scanning efficiency $> 97\%$) and no associated neutral hadron interactions or neutral strange particle decays. Cuts were made on the energy and angle of the charged decay products to ensure consistency with the assumed production/decay channel.

One then needs a decay distribution as input to calculate the expected cut efficiency. In the three-body leptonic decays, the energy of the children fully specifies the relative orientation of the 3-momenta, so all that is required is the differential distribution for the two energies $E, E'$. In the HNL rest frame, this is:

$$\frac{\mathrm{d}^2 \Gamma}{\mathrm{d}E \mathrm{d}E'} = \frac{G_F^2 m_N}{2\pi^3} \left[ g_-^2 E(m_N - 2E) + g_+^2 E'(m_N - 2E') + 4m^2 g_- g_+ \left( 1 - \frac{E + E'}{m_N} \right) \right], \tag{12}$$

where $g_- = \sin^2 \theta_{\mathrm{W}}$ and $g_+ = (\sin^2 \theta_{\mathrm{W}} - \frac{1}{2})^2$, and we keep explicit finite mass corrections. This formula applies to all decays via the neutral current with massive final state particles

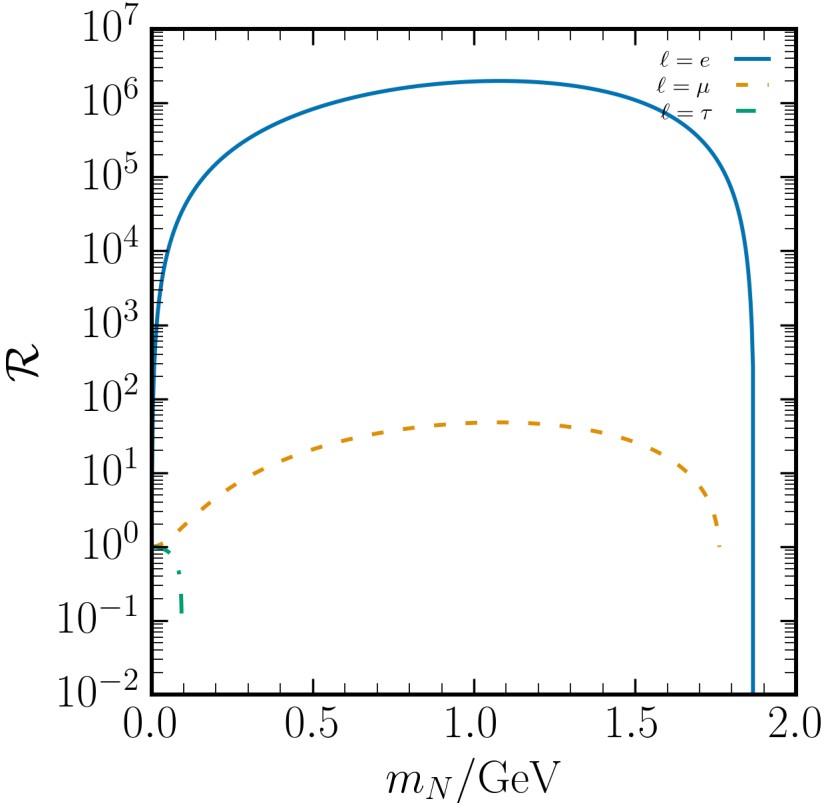

Figure 3: The enhancement of HNL production relative to massless neutrinos in two-body meson decays (8). For HNLs mixing with the electron or muon neutrino, there is significant enhancement due to relaxation of the helicity suppression as the HNL mass increases. However for HNLs mixing with the tau neutrino the enhancement is negligible; this explains why no bound was quoted in the previous BEBC analysis [23].

and agrees with e.g. the massless $m^2 \to 0$ limit computed in [48]. The hadronic decays, meanwhile, are two-body ($\not{p}_T \equiv -p_T = 0$), so the distribution is trivial.

Since the sensitivity to HNLs depends on the experimental cuts that were used to isolate signal events, we must take all of this into account to extract bounds on the HNL mixing angles. In particular, we require that HNL events pass a cut on the invariant transverse mass $M_T$, defined by

$$M_T \equiv (p_T^2 + M_I^2)^{1/2} + p_T < m_D - m_\mu. \qquad (13)$$

We further adopt a lepton identification efficiency of 96%. This was the detection efficiency of the WA66 experiment for electron tracks of momentum $> 0.8\,\mathrm{GeV}/c$, while it was 97% for muons of momentum $> 3\,\mathrm{GeV}/c$ [23]. (While the cuts used depended on the HNL under consideration, no specific results for mixing with $\nu_\tau$s were given, hence we conservatively use the same cuts that were placed on HNLs mixing with $\nu_\mu$s.)

There were no surviving candidates in WA66 for the HNL decay channels $ee\nu$, $e\mu\nu$ or $\mu\mu\nu$, or for $e\pi$, and there was only 1 candidate for $\mu^+\pi^-$ (with invariant mass $\sim$ 1 GeV). The background for this decay channel was estimated using data from the WA59 experiment [49] in which BEBC, filled with a $\mathrm{Ne}/\mathrm{H}_2$ mix similar to WA66, was exposed to a conventional 'wide band' beam (in which the fraction of HNLs would have been $< 1\%$ of that in the beam dump beam). This background was $0.6 \pm 0.2$ events [23] corresponding to an upper limit of 3.5 events @ 90% CL with one candidate event. Since there were

| Experiment | POT/$10^{18}$ | $E_{\rm b}$/GeV | $D$/m | $V$/cm$^3$ | Cuts | Observed events (Background) | $\eta$ |
|---|---|---|---|---|---|---|---|
| BEBC [22, 23] | 2.72 | 400 | 404 | $357 \times 252 \times 185$ | $E_{\rm T} > 1\,{\rm GeV}$ $\wedge M_{\rm T} < m_D - m_\mu$ | 1 $\mu^+\pi^-$ (0.6±0.2) | 0.96 |

Table 1: The relevant experimental parameters for the CERN-WA66 experiment. POT is the total number of protons on target, $E_{\rm b}$ is the energy of proton beam, $D$ is the distance from the end of the target to the beginning of the detector and $V$ is the detector volume written as transverse area $\times$ length; the dimensions of BEBC are given approximating the detector as a cuboid. Cuts are placed on the total energy of the charged pair $E_{\rm T}$ and on the transverse mass, as defined in Eq.(13). The number of observed events is given, as well as the estimated background. The detection efficiency after cuts is denoted as $\eta$.

no candidate events in the 3-body channels available to $U_{eN}$ or $U_{\tau N}$ mixing, we have conservatively adopted an upper limit of 2.3 signal events [50].

## 3.1 Decay rates

We now evaluate the partial and total decay widths $\Gamma_\alpha$ and $\Gamma$ in Eq.(11). The HNL is taken to be a Majorana fermion. As with HNL production, the phenomenology depends on the HNL mass and the mixing parameters. (For the mixing with $\nu_\tau$ the total width is calculated below. For the mixing with $\nu_e$ there are additional contributions [43].)

The total width is dominated by hadronic decays, once these are kinematically allowed as detailed in [43]. Below the QCD scale, the width is dominated by the decay to a neutral pion at a rate

$$\Gamma(N \to \nu_\alpha \pi^0) = \frac{G_F^2 f_\pi^2 m_N^3}{32\pi} |U_{\alpha N}|^2 \left(1 - x_{\pi N}\right)^2, \tag{14}$$

with $x_{\pi N} \equiv (m_\pi/m_N)^2$. Above $\Lambda_{\rm QCD}$ there are significant contributions from multi-hadron final states which we approximate by the decay width to quarks:

$$\Gamma(N \to \nu_\alpha f\bar{f}) = \frac{G_F^2 m_N^5}{192\pi^3} |U_{\alpha N}|^2 c_f, \tag{15}$$

where the constants are $c_u = 3(1 - \frac{8}{3}\sin^2\theta_{\rm W} + \frac{32}{9}\sin^4\theta_{\rm W})/4$ and $c_d = 3(1 - \frac{4}{3}\sin^2\theta_{\rm W} + \frac{8}{9}\sin^4\theta_{\rm W})/4$. We augment this with a QCD loop factor, which we take to be the same as in the corresponding tau decay [51].

The decay to a lepton pair is also described by Eq.(15) in the limit when the lepton pair is much lighter than the HNL. In this case, the coefficient $c_f$ depends on whether there is a charged current contribution to the rate in addition to the neutral current contribution. When there is only a neutral current contribution, $c_\ell = (1 - 4\sin^2\theta_{\rm W} + 8\sin^4\theta_{\rm W})/4 \simeq 0.13$, while $c_\ell = (1 + 4\sin^2\theta_{\rm W} + 8\sin^4\theta_{\rm W})/4 \simeq 0.57$ if both currents contribute [13]. The bounds quoted earlier in [23] use $c_\ell = 1$, which corresponds to a charged current-only interaction.

## 4 Results and conclusions

Fig. 4 shows our bound on $U_{\tau N}$. Remarkably, BEBC WA66 outperforms all other experiments, including the much bigger CHARM detector. This is primarily because its decay region was off-axis to the beam so it had a lower geometric acceptance than BEBC, as well as receiving a smaller fraction of high energy HNLs. Consequently the on-axis BEBC sets a tighter bound as the HNL mass increases and the transverse momentum gets smaller.

However once the HNL mass exceeds $m_\tau$, there are no limits from the old fixed target experiments where sufficient numbers of $B$ mesons were not produced. This will happen however at the high luminosity LHC where experiments at the FPF will probe HNLs with mass up to $m_B$ [5], as well as at future lepton colliders [52, 53]. In the interim, new searches for GeV mass HNLs will be carried out with the LHCb upgrade [54], and NA62 in beam dump mode [55] as well as FASER 2 [56, 57], with proposed experiments such as CODEX-B, MATHUSLA and SHiP to hopefully follow [28, 29].

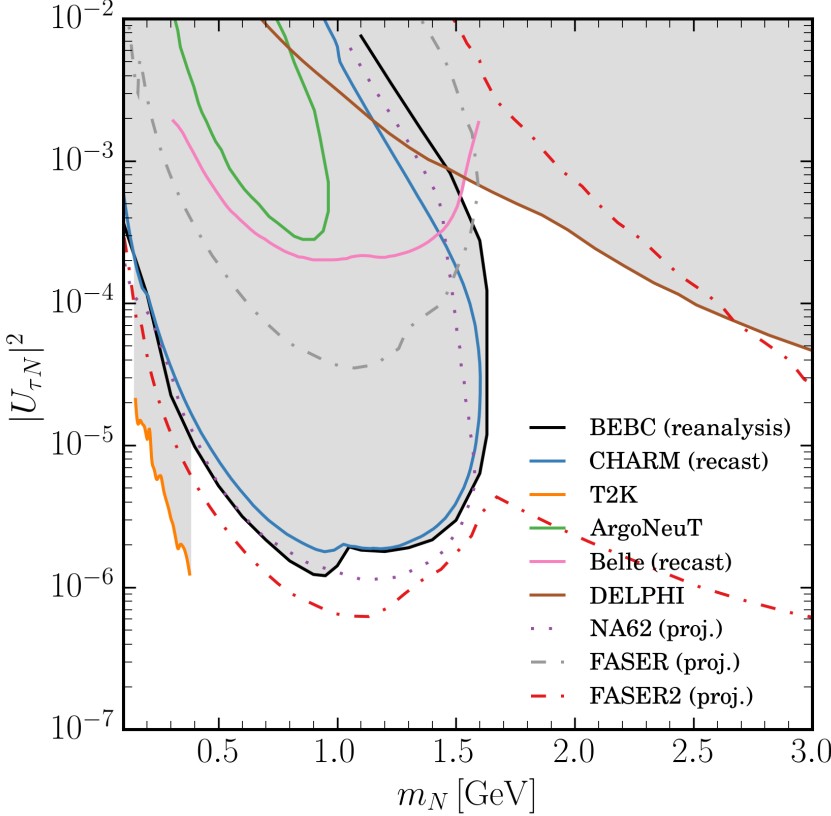

Figure 4: The 90% exclusion region in the HNL mass versus its mixing with $\nu_\tau$ set by reanalysis of BEBC WA66, compared to the recast [13] of the CHARM bound. Also shown are bounds from T2K [6], ArgoNeuT [7], a recast of Belle [8], and DELPHI [15], as well as the projected sensitivities of NA62 in beam dump mode [55] and FASER/FASER2 [57].

We also show in Fig. 5 updated bounds from BEBC WA66 [23] on $U_{eN}$, the mixing with the electron neutrino. Using a corrected formula for the HNL decay probabilities, additional production channels, as well as an improved fit for the $D$ meson distribution results in a two-fold improvement over the bounds previously obtained. Note that the widely quoted bound from CHARM [14] was shown as extending up to HNL mass of 2.2 GeV, which is well beyond the kinematic limit. Fig. 5 shows the corrected and updated version [13] of this bound .

We have demonstrated the continued capability of the BEBC detector to place world-leading bounds on hypothetical particles of interest. This reanalysis has taken into account production and decay channels of HNLs with non-zero $\nu_\tau$ mixings that have not been much considered earlier, thus providing an up-to-date set of exclusions. It would be interesting to explore the sensitivity of BEBC to other models of HNLs, for instance those involving new mediators [46, 47], as we expect similar improvements may be had.

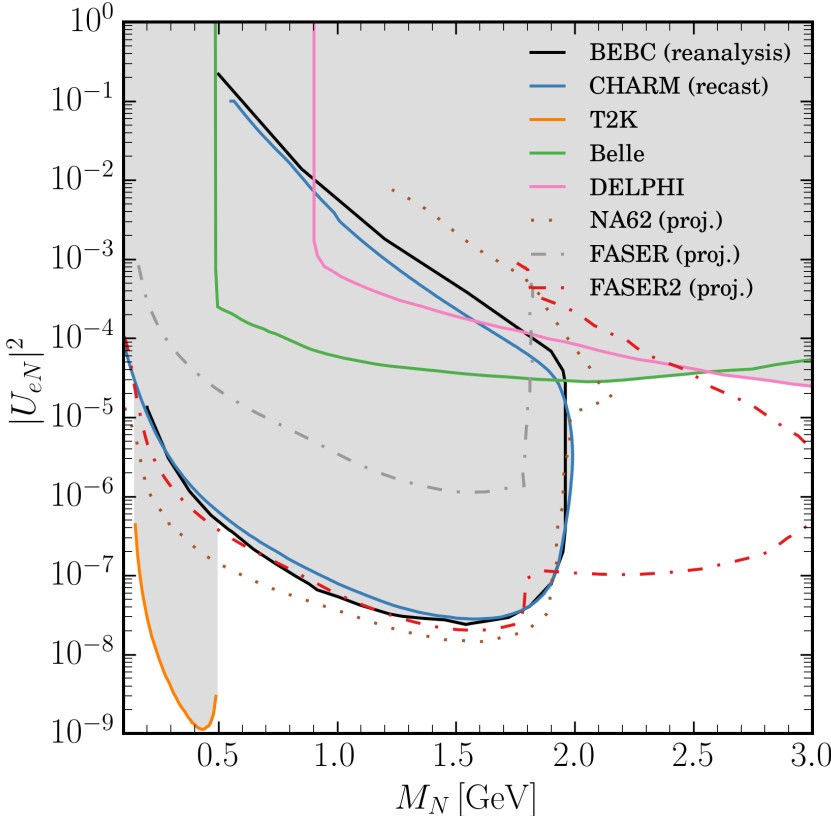

Figure 5: The 90% exclusion region in the HNL mass versus its mixing with $\nu_e$ set by this reanalysis of BEBC WA66. The recast bound [13] from CHARM is also shown, as are bounds from T2K [6], Belle [10], and DELPHI [15], as well as the projected sensitivities of NA62 in beam dump mode [55] and FASER/FASER2 [57].

# Acknowledgments

We are grateful to the late Per-Olof Hulth, and all those who worked on the CERN-WA-066 BEBC beam dump experiment, for having done such a great job that it continues to provide us such rich dividends 40 years later. We thank Suchita Kulkarni, Maksym Ovchynnikov and Sonali Verma for helpful correspondence, and Amanda Cooper-Sarkar and Wilbur Venus for discussions.

**Code** The results in this paper may be reproduced using the python scripts available at: `https://github.com/ryanbarouki/HNL_Dump`.

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
