# Peer review of "Blast from the past II: Constraints on heavy neutral leptons from the BEBC WA66 beam dump experiment"

_SciPost Physics_

## Round 2 · Referee Report · Anonymous (Referee 1) · 2022-9-1

Report
The paper presents a recast of the past bounds from the BEBC experiment on the parameter space of the beyond the Standard Model scenario with GeV-scale heavy neutral leptons (HNLs). The main focus of the paper is on the dominant mixing with the tau neutrino while the relevant recast for the electron mixing is also provided.
The model under study is very popular, in fact it is one of a few benchmark scenarios routinely used in assessing the expected sensitivity of new experimental proposals in this field of particle physics. Hence, presenting the updated bounds on this scenario remains timely and interesting, especially given that the “new bounds” found in the paper happen to be the leading constraints on this scenario in a certain region of the parameter space of the model.
I generally find the discussion contained in the draft to be clear and worth-publishing. I only have minor remarks to the manuscript as follows. Since the denominator in eq. (2) should correspond to the measured number of active neutrinos, the cross sections present on the RHS are not the total ones but they rather already implicitly take into account the angular acceptance of the detector and other cuts on the resulting neutrino interactions. It would be useful to the reader to clarify the following points in the discussion of this equation:
1) Are the simple relations between the different meson production cross sections inferred from the Fermilab E769 experiment and mentioned in the text still valid for the BEBC setup?
2) Given that HNLs for the tau mixing are dominantly produced in tau lepton decays, which corresponds to an additional step in the decay chain from charm mesons, one would expect that the parent charm meson spectrum in this case would corresponds to somewhat different angular and energy spectrum than the one for the active neutrino production, i.e. the relevant cross section \sigma in the numerator of eq. (2) would be a somewhat different quantity. Could this influence the cancellation of hadronic uncertainties mentioned in the discussion below eq. (2)?
3) Is the active tau neutrino production in tau lepton decays irrelevant for the denominator of eq. (2)?

Author: Giacomo Marocco on 2022-09-08 [id 2796]
(in reply to Report 1 on 2022-09-01)Thank you for the thoughtful report and questions. Our responses are below:
The denominator of equation (2) corresponds to the total number of active neutrinos, rather than the measured number, so acceptances, cuts, etc. are all done after. We shall emphasise this in the text, and modify any ambiguous statements.
1) As mentioned, equation (2) corresponds to total cross-sections, and so modulo energy and target-composition dependence, these simple relations ought to hold.
2) The tau leptons arise from charmed meson decay, as the referee correctly states. As far as equation 2 is concerned, the only modification introduced by the additional step in the decay chain is that one must multiply any charmed meson cross-sections by a branching ratio corresponding to the decay to a tau lepton. The fact that these tau leptons will have different differential distributions to their parents is incorporated when we calculate the acceptances using our MC. In this process, we assume that all the charmed mesons all follow the same differential distributions (due to a lack of D_s spectral data), as mentioned in the text.
3) The denominator of equation (2) corresponds to the production channels of a specific flavour of active neutrino \nu_\ell. We will clarify this in the text, and also correct a typo to specify that the 4.1 × 10−4 neutrinos produced via D decays per POT are all muon neutrinos.
Anonymous on 2022-09-21 [id 2838]
(in reply to Giacomo Marocco on 2022-09-08 [id 2796])Thank you, this sounds fair enough.

---

## Round 2 · Referee Report · Anonymous (Referee 2) · 2022-9-12

Report
This paper discusses the constraint from an early beam dump experiment on the heavy neutral lepton (HNL). The HNL is a well motivated candidate for new physics in neutrino and dark matter models. It is under scrutiny in recent phenomenological and experimental works. Obtaining a new limit in its parameter space is an important result.
I have two comments for the authors to consider before publishing.
1) At the beginning of the paper, the author mentioned inverse seesaw mechanism as the theoretical context for HNL. In fact, a HNL with a mixing with the active neutrino of interest to this work, could occur in a broader class of theories, such as the type-I seesaw mechanism, as well as neutrino portal dark sector models where the HNL is a Dirac fermion. It will be worth commenting on these possibilities.
2) In the resulting Figures 4 and 5, I suggest the authors to show the reach of a few upcoming experiments such as FASER2 and NA62. Although they are commented in the main text, it would be useful to see how the BEBC limit found here compare with the future probes.

---

## Editorial Decision

unknown